



# Amino acid racemization in *Neogloboquadrina pachyderma* and *Cibicidoides wuellerstorfi* from the Arctic Ocean and its implications for age models

Gabriel West[1,2], Darrell S. Kaufman[3], Martin Jakobsson[1,2], Matt O'Regan[1,2]

[1] Department of Geological Sciences, Stockholm University, Stockholm, 10691, Sweden
[2] Bolin Centre for Climate Research, Stockholm University, Stockholm, 10691, Sweden
[3] School of Earth and Sustainability, Northern Arizona University, Flagstaff, AZ 86011, USA

*Correspondence to*:  Gabriel West (gabriel.west@geo.su.se)
                       Matt O'Regan (matt.oregan@geo.su.se)

**Abstract**

We report the results of amino acid racemization (AAR) analyses of aspartic and glutamic acids in the planktic foraminifera, *Neogloboquadrina pachyderma*, and the benthic species, *Cibicidoides wuellerstorfi*, collected from sediment cores from the Arctic Ocean. The cores were retrieved at various deep-sea sites of the Arctic, which cover a large geographical area; from the Greenland and Iceland seas to the Alpha and Lomonosov Ridges in the

central Arctic Ocean. Age models for the investigated sediments were developed by multiple dating techniques, including oxygen isotope stratigraphy, magnetostratigraphy, bio-, litho-, and cyclostratigraphy. The extent of racemization (D/L values) was determined on 95 samples (1028 subsamples) and shows a progressive increase downcore for both foraminifera species. Differences in the rates of racemization between the species were established by analysing specimens of both species from the same stratigraphic levels (n = 21). Aspartic acid and

glutamic acid racemize on average 16±2 % and 23±3 % faster, respectively, in *C. wuellerstorfi* than in *N. pachyderma*. D/L values typically increase with sample age, with a trend that follows a simple power function. Scatter around least square regression fits are larger for samples from the central Arctic Ocean than for those from the Nordic Seas. Calibrating the rate of racemization in *C. wuellerstorfi* using independently dated samples from the Greenland and Iceland seas for the past 400 ka enables estimation of sample ages from the central Arctic

Ocean, where bottom water temperatures are similar. The resulting ages are older than expected when considering the existing age models for the central Arctic Ocean cores. These results confirm that the differences are not due to taxonomic effects and further warrant a critical evaluation of existing Arctic Ocean age models and the environmental factors that may influence racemisation rates in central Arctic Ocean sediments.






## 1. Introduction

The first application of amino acid geochronology to Arctic Ocean sediments analysed the extent of racemization in the protein amino acid, isoleucine over time in samples of the planktic foraminifera, *Neogloboquadrina pachyderma* and the benthic species, *Cibicidoides wuellerstorfi* (Sejrup et al., 1984). Not only did this study provide some of the first amino acid racemization (AAR) data from a polar environment, but it also exposed crucial chronological issues associated with Arctic sediments. The results contradicted available age

interpretations obtained from palaeomagnetic data (Sejrup et al., 1984; Backman, 2004). The problems of dating Pleistocene Arctic marine sediments continue to exist today and are well known (e.g. Alexanderson et al., 2014). Over the past few decades, amino acid geochronology received limited attention in the Arctic, but several studies provided promising results (Sejrup & Haugen, 1992; Kaufman et al., 2008, 2013) that highlighted its potential as a dating technique, and the need for its continued development in Arctic settings. This is particularly desirable,

since theoretically, it could provide age control up to a few million years, using even limited amounts of biocarbonate.

   *N. pachyderma* and *C. wuellerstorfi* are commonly used in stable isotope stratigraphy and paleoceanographic reconstructions (e.g. Shackleton et al., 2003) and are associated with cold water masses. *N. pachyderma* is

considered to be a primarily 'high-latitude' species (Darling et al., 2017), and *C. wuellerstorfi* is thought to show strong preference for bottom waters below ~5ºC (Rasmussen and Thomsen, 2017). These characteristics and their frequent occurrence in sediment cores from the Arctic Ocean make them particularly useful for amino acid geochronology studies in this region.

The rates of racemization for aspartic and glutamic acids were previously calibrated in *N. pachyderma* from central Arctic Ocean samples by Kaufman et al. (2008) for the past 150 ka. The calibration relied on the established age of upper Quaternary sediments from the Lomonosov Ridge (O'Regan et al., 2008). Subsequently, however, the extent of racemization in these samples was shown to be higher than expected when compared with those of similar ages from other cold bottom water sites from the Atlantic and Pacific oceans (Kaufman et al., 2013). The

reasons for this apparently higher extent of racemization in *N. pachyderma* from central Arctic Ocean samples is unclear, but not considered to be caused by taxonomic effects, since the rate of racemization is lower in this species than in other observed taxa (Kaufman et al., 2013). Either the established ages that were used to calibrate the rate of racemization in the central Arctic Ocean sediments are too young, such that units currently correlated with substages of marine oxygen isotope stage (MIS) 5 instead represent MIS 9, 7 and 5, or other – undetermined

– processes influence protein degradation and preservation in the central Arctic Ocean. At the Yermak Plateau (Fig. 1), racemization rates for *N. pachyderma* generally conform to the rates determined for other cold bottom water sites (West et al., 2019), further challenging the established ages previously used to calibrate the rate of AAR from the Lomonosov Ridge. However, it is unknown how racemization progresses in other regions of the Arctic Ocean.


   If the apparently higher extent of racemization in *N. pachyderma* from the central Arctic Ocean is not the result of taxonomic effects, a higher rate of racemization can also be anticipated in other taxa from the area, e.g. in




*Cibicidoides wuellerstorfi.* However, little is known about racemization of amino acids in this species. The earliest studies involving *C. wuellerstorfi* investigated taxonomical applications (Haugen et al., 1989), or focused on the epimerization of isoleucine (Sejrup et al., 1984; Sejrup and Haugen, 1992) utilising HPLC ion exchange analysers. Since the publication of these seminal papers, analysis of amino acids has become significantly faster, with reduced sample size requirements, due to improvements in analytical methods (Kaufman & Manley, 1998), yet no studies have addressed amino acid racemization in *C. wuellerstorfi* despite its palaeoceanographical importance (e.g. Yu & Elderfield, 2008; Wollenburg et al., 2015; Burkett et al., 2016; Raitzsch et al., 2020), and the relatively faster and easier sample processing offered by its larger tests (up to ~4-5 times the size) when compared to *N. pachyderma*.

Here we report the results of aspartic acid and glutamic acid racemization analyses of *Neogloboquadrina pachyderma* and *Cibicidoides wuellerstorfi* obtained from well-dated Quaternary deep-sea sediment cores from the Greenland and Iceland seas, and from sediment cores from the central Arctic Ocean, where sediment ages are less certain. The long-term rates of racemization in the two species are compared, and the relationship between the extent of racemization and sample age is investigated in both species.

## 2. Materials and methods

### a) Investigated sediment cores

Foraminifera samples were taken from sediment cores from the Greenland and Iceland seas, the Lomonosov Ridge, and the Alpha Ridge (Fig. 1). The Greenland and Iceland seas, although part of the Arctic Ocean, are under the direct influence of the Atlantic, and thus are predominantly characterised by open water conditions, unlike the sea ice covered areas of the Lomonosov and Alpha Ridges. The studied sediment cores (Table 1) were collected from deep water (811 – 2952 m) environments, which – based on modern estimates – experience similar, very cold (<0 ºC), and relatively stable bottom water temperatures.

| Region | Core | Latitude (º) | Longitude (º) | Water depth (m) | Bottom water temperature est. (º) |
|---|---|---|---|---|---|
| Lomonosov Ridge | AO16-5-PC1 | 89.0780 | -130.5470 | 1253 | -0.3 |
| Alpha Ridge | AO16-9-PC1 | 85.9557 | -148.3258 | 2212 | -0.4 |
| Lomonosov Ridge | LOMROG07-PC04 | 86.7012 | -53.7672 | 811 | 0.0 |
| Lomonosov Ridge | LOMROG12-PC03 | 87.7247 | -54.4253 | 1607 | -0.4 |
| Lomonosov Ridge | LOMROG12-PC07 | 88.1976 | -55.6845 | 2952 | -0.8 |
| Lomonosov Ridge | LOMROG12-PC09 | 89.0267 | -73.7344 | 1318 | -0.3 |
| Iceland Sea | ODP 151/ 907A | 69.2498 | -12.6982 | 1801 | -0.8 |
| Greenland Sea | PS17 / 1906-2 | 76.8463 | -2.1505 | 2901 | -1.0 |

**Table 1: Sediment cores investigated in this study. Current bottom water temperatures were approximated by using annual mean temperature observations from the nearest location from the World Ocean Atlas (Locarnini et al., 2018).**



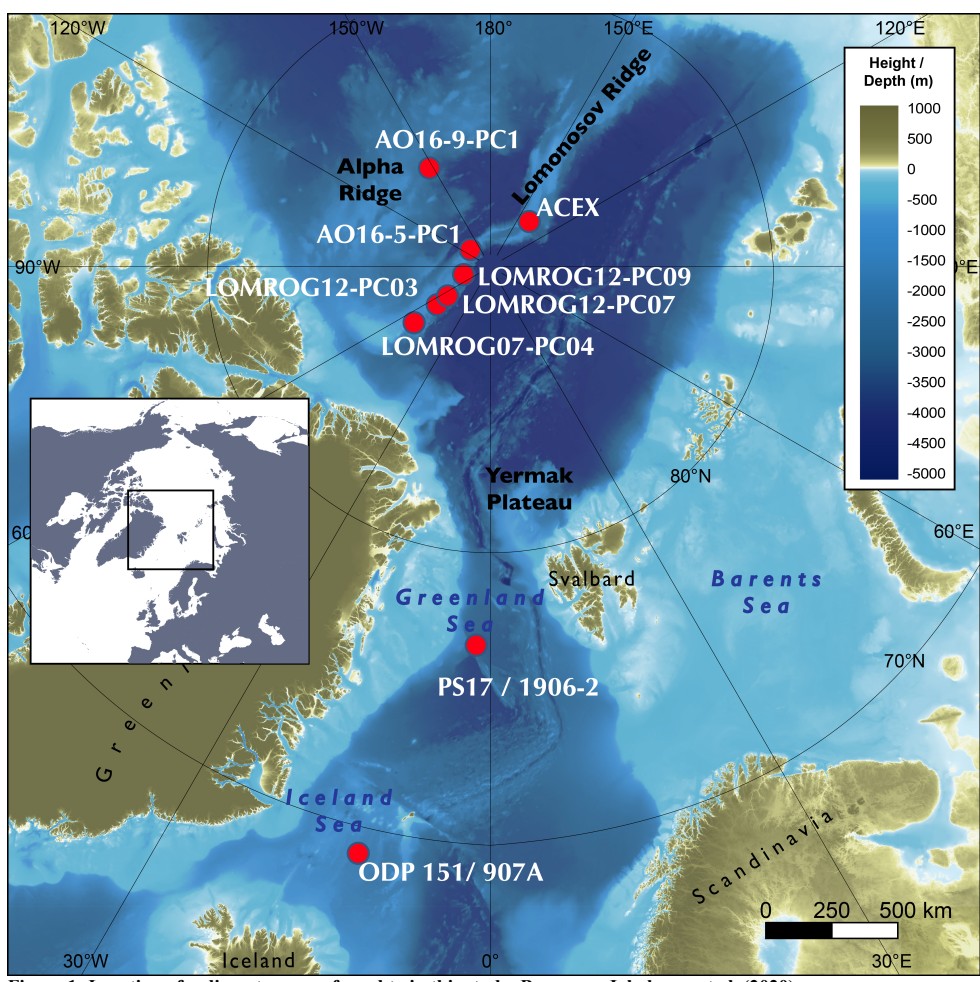

**Figure 1: Location of sediment cores referred to in this study. Basemap: Jakobsson et al. (2020).**

Age-depth models for the investigated sediment cores were developed using a variety of dating techniques. Cores from the Nordic Seas primarily relied on oxygen isotope stratigraphy, complemented by magnetostratigraphy in the case of ODP151/907A (Jansen et al., 2000a, b), and by carbon isotope stratigraphy for core PS1906-2 (Bauch, 2002, 2013). The age-depth model of the latter is less certain beyond marine isotope stage (MIS) 6 (Bauch, 2013), due to the large uncertainty associated with isotope stratigraphy, a characteristic issue of Arctic Ocean records.


The age-depth models of sediment cores from the central Arctic Ocean utilise a more diverse toolset, reflecting the difficulties of dating Arctic marine sediments, and heavily depend on a combination of bio- and lithostratigraphy (e.g. Cronin et al., 2019). The lithostratigraphy of the central Arctic Ocean cores investigated in this study can be correlated to that of the Integrated Ocean Drilling Program Expedition 302, the Arctic Coring

Expedition (ACEX). This correlation is most apparent when bulk density profiles are used (Fig. 2), but it has been





shown to be also coherent when other sedimentological properties including grain size and a variety of XRF-scanning properties (O'Regan et al., 2019) are utilised. The currently accepted age model for the ACEX sedimentary sequence was developed using cyclostratigraphic analysis (O'Regan et al., 2008) and produced similar estimated Quaternary sedimentation rates as obtained by the decay of beryllium isotopes (Frank et al.,

2008). The late Quaternary chronology (MIS 1 – 6) for ACEX included constraints from [14]C dating, the correlation to near-by records AO96/12-1PC (Jakobsson et al., 2001) and PS2185 (Spielhagen et al., 2004), where MIS 5 was identified based on the occurrence of the calcareous nannofossil *Emiliania huxleyi* (Jakobsson et al., 2001), and further supported by results from optically stimulated luminescence dating of quartz grains (Jakobsson et al., 2003). The age model of core LOMROG07-PC04 is based on correlation to PS2185 (Hanslik et al., 2013).


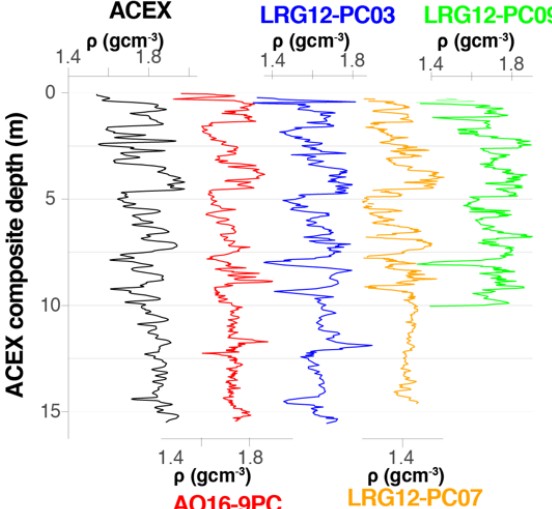

**Figure 2: Correlation of bulk density (ρ) profiles of sediment cores from the central Arctic Ocean. Depth profiles were all scaled linearly to match the corresponding depths in the Arctic Coring Expedition (ACEX) core.**

b)   _Analytical procedures_

Sediment samples were wet sieved (63 μm), and air dried prior to picking foraminifera tests of *N. pachyderma* and *C. wuellerstorfi*. Initially, the over-250-μm fraction was targeted to isolate the largest and best-preserved tests. For some samples this was not possible, and the tests were collected from the 180 – 250 μm fraction instead. The

tests were kept in glass vials, and stored in a refrigerator prior to racemization analyses. A total of 95 stratigraphic depths were sampled, with some depths containing both foraminifera species. Each sample was further subsampled – on average with 9.6 *N. pachyderma* and 11.2 *C. wuellerstorfi* subsamples per sample, producing 1028 analysed subsamples. Each subsample comprised between 10 and 12 *N. pachyderma*, or 2 and 4 *C. wuellerstorfi* tests.


The analytical procedures followed that of previous analyses as described in detail by Kaufman et al. (2013) and West et al. (2019), and were performed at the Amino Acid Geochronology Laboratory (AAGL), Northern Arizona



University. The foraminifera tests were first sonicated (1-30 s) to remove any loose sediment particles, treated with 1 ml hydrogen peroxide (3%) to remove surficial organic matter, and then rinsed three times with reagent

grade (grade I) water. Tests were picked into micro-reaction hydrolysis vials (defining one subsample), and 8 µl hydrochloric acid (6 M) was added to dissolve the tests. The vials were then sealed with nitrogen gas, and the subsamples hydrolysed for 6 hours at 110ºC. After the hydrolysis was complete, the subsamples were evaporated in a vacuum desiccator, and then rehydrated in 4 µl of 0.01 M HCl spiked with 10 µM L-homoarginine. Each subsample was injected onto a high-performance liquid chromatograph (HPLC) with a fully automated, reversed

phase procedure (Kaufman & Manley, 1998) to separate pairs of D- and L-amino acids. The peak-area ratio of D and L stereoisomers of eight amino acids (aspartic acid, glutamic acid, serine, alanine, valine, phenylalanine, isoleucine and leucine) were analysed to determine the extent of racemization, but this study only utilised the results of aspartic acid (Asp) and glutamic (Glu) acid racemization, among the two most abundant and chronographically well-resolved amino acids.


### 3. Results

### a) Data screening

Initial data screening was based on the procedure of Kosnik & Kaufman (2008). First, subsamples with L-Ser / L-Asp ≥ 0.8 – an indicator of potential contamination by modern amino acids – were excluded. The D/L values of Asp and Glu positively covary in fossil proteins. Subsamples, which did not adhere to this expected trend were omitted. Finally, subsamples with D/L Asp or Glu values not within ±2σ of the sample mean were also removed (Supplementary Fig. S1). As a result of this screening process, 16.0 % of all subsamples were rejected

(Supplementary Table S1).

Following the subsample screening process, sample means, and related standard deviation values were calculated for Asp and Glu for all samples. Stratigraphically reversed samples (mean D/L values lower than expected for their stratigraphic depths with no overlap within 1σ with the sample from shallower depth) were identified within

each core (Table 2).

| UAL | Core | Core depth (m) | Age (ka) | $n^a$ | excl. | excl. ratio (%) | Asp D/L | 1 σ | Glu D/L | 1 σ |
|---|---|---|---|---|---|---|---|---|---|---|
| *Neogloboquadrina pachyderma* | | | | | | | | | | |
| | **Alpha Ridge** | | | | | | | | | |
| 17336 | AO16-9-PC1 | 0.190 | 16 | 10 | 1 | 10 | 0.141 | 0.014 | 0.069 | 0.011 |
| 17337 | AO16-9-PC1 | 0.210 | 22 | 10 | 1 | 10 | 0.210 | 0.010 | 0.104 | 0.005 |
| 17338 | AO16-9-PC1 | 0.970 | 127 | 9 | 4 | 44 | 0.419 | 0.020 | 0.242 | 0.029 |
| 17339 | AO16-9-PC1 | 1.260 | 188 | 10 | 2 | 20 | 0.386 | 0.090 | 0.212 | 0.062 |
| | **Lomonosov Ridge** | | | | | | | | | |
| 22759 | LOMROG07-PC04 | 0.025 | MIS 1-3? (~19) | 17 | 1 | 6 | 0.113 | 0.029 | 0.050 | 0.015 |
| 21587 | LOMROG07-PC04 | 0.040 | MIS 1-3? (~39) | 8 | 2 | 25 | 0.105 | 0.022 | 0.038 | 0.007 |
| 21588 | LOMROG07-PC04 | 0.140 | MIS 5.1? (~77) | 8 | 2 | 25 | 0.294 | 0.049 | 0.132 | 0.046 |
| 22760 | LOMROG07-PC04 | 0.225 | MIS 5.1? (~82) | 11 | 1 | 9 | 0.278 | 0.008 | 0.123 | 0.009 |
| 22761 | LOMROG07-PC04 | 1.025 | MIS 5.5? (~115) | 13 | 1 | 8 | 0.322 | 0.013 | 0.153 | 0.013 |
| 21589 | LOMROG07-PC04 | 1.110 | MIS 5.5? (~123) | 8 | 4 | 50 | 0.347 | 0.029 | 0.146 | 0.022 |
| 22762 | LOMROG07-PC04 | 1.385 | > MIS 7? (~405) | 13 | 2 | 15 | 0.387 | 0.029 | 0.195 | 0.032 |





| ID | Core | | | | | | | | | |
|---|---|---|---|---|---|---|---|---|---|---|
| [b]21590 | LOMROG07-PC04 | 1.410 | MIS 1-3? (~12.8) | 8 | 3 | 38 | 0.318 | 0.035 | 0.137 | 0.035 |
| 21591 | LOMROG07-PC04 | 1.720 | MIS 1-3? (~40) | 8 | 3 | 38 | 0.391 | 0.036 | 0.175 | 0.043 |
| 22763 | LOMROG07-PC04 | 1.785 | MIS 5.1? (~74) | 12 | 1 | 8 | 0.371 | 0.049 | 0.177 | 0.039 |
| 21592 | LOMROG07-PC04 | 2.010 | MIS 5.1? (~78) | 8 | 7 | 88 | 0.357 | | 0.140 | |
| 22764 | LOMROG07-PC04 | 2.025 | MIS 5.5? (~115) | 9 | 1 | 11 | 0.401 | 0.079 | 0.220 | 0.067 |
| [b]21594 | LOMROG07-PC04 | 3.000 | MIS 5.5? (~123) | 7 | 4 | 57 | 0.270 | 0.099 | 0.136 | 0.057 |
| 21408 | LOMROG12-PC03 | 0.505 | 61 | 8 | 2 | 25 | 0.340 | 0.032 | 0.163 | 0.026 |
| 21409 | LOMROG12-PC03 | 0.610 | 76 | 8 | 1 | 13 | 0.395 | 0.021 | 0.206 | 0.015 |
| 21410 | LOMROG12-PC03 | 0.761 | 108 | 8 | 3 | 38 | 0.415 | 0.040 | 0.212 | 0.025 |
| 21411 | LOMROG12-PC03 | 1.291 | 204 | 8 | 2 | 25 | 0.468 | 0.015 | 0.240 | 0.010 |
| 21412 | LOMROG12-PC03 | 1.401 | 236 | 8 | 1 | 13 | 0.431 | 0.092 | 0.236 | 0.078 |
| 21413 | LOMROG12-PC03 | 1.551 | 290 | 8 | 4 | 50 | 0.466 | 0.066 | 0.242 | 0.070 |
| 21414 | LOMROG12-PC03 | 1.601 | 311 | 9 | 3 | 33 | 0.431 | 0.044 | 0.203 | 0.050 |
| 21415 | LOMROG12-PC03 | 1.781 | 391 | 6 | 0 | 0 | 0.401 | 0.081 | 0.208 | 0.060 |
| 21416 | LOMROG12-PC03 | 2.101 | 489 | 9 | 5 | 56 | 0.422 | 0.067 | 0.238 | 0.069 |
| 15866 | LOMROG12-PC07 | 0.020 | 9 | 9 | 2 | 22 | 0.081 | 0.004 | 0.034 | 0.002 |
| 15867 | LOMROG12-PC07 | 0.165 | 31 | 10 | 1 | 10 | 0.205 | 0.036 | 0.090 | 0.022 |
| 15868 | LOMROG12-PC07 | 0.315 | 40 | 10 | 2 | 20 | 0.272 | 0.025 | 0.124 | 0.013 |
| 15869 | LOMROG12-PC07 | 0.780 | 52 | 10 | 2 | 20 | 0.294 | 0.026 | 0.122 | 0.019 |
| 15870 | LOMROG12-PC07 | 0.980 | 63 | 10 | 5 | 50 | 0.354 | 0.024 | 0.163 | 0.022 |
| 15871 | LOMROG12-PC07 | 1.170 | 79 | 9 | 2 | 22 | 0.355 | 0.053 | 0.183 | 0.030 |
| 15872 | LOMROG12-PC07 | 1.425 | 96 | 9 | 3 | 33 | 0.399 | 0.035 | 0.183 | 0.027 |
| 15873 | LOMROG12-PC07 | 1.580 | 111 | 9 | 2 | 22 | 0.368 | 0.030 | 0.168 | 0.024 |
| 15874 | LOMROG12-PC07 | 1.720 | 124 | 9 | 3 | 33 | 0.385 | 0.024 | 0.173 | 0.025 |
| [b]15875 | LOMROG12-PC07 | 3.537 | 350 | 9 | 3 | 33 | 0.305 | 0.036 | 0.133 | 0.018 |
| 15876 | LOMROG12-PC07 | 3.657 | 374 | 9 | 0 | 0 | 0.354 | 0.048 | 0.185 | 0.031 |
| 15877 | LOMROG12-PC07 | 3.787 | 402 | 5 | 4 | 80 | 0.348 | | 0.174 | |
| 15878 | LOMROG12-PC07 | 4.210 | 485 | 7 | 3 | 43 | 0.390 | 0.027 | 0.185 | 0.019 |
| 15879 | LOMROG12-PC07 | 4.405 | 502 | 7 | 1 | 14 | 0.348 | 0.032 | 0.165 | 0.027 |
| [b]15880 | LOMROG12-PC07 | 4.510 | 510 | 5 | 3 | 60 | 0.307 | 0.029 | 0.145 | 0.034 |
| 22765 | LOMROG12-PC09 | 0.800 | 55 | 17 | 1 | 6 | 0.378 | 0.013 | 0.195 | 0.012 |
| 22766 | LOMROG12-PC09 | 1.250 | 129 | 10 | 0 | 0 | 0.400 | 0.029 | 0.209 | 0.030 |
| 22767 | LOMROG12-PC09 | 1.840 | 205 | 12 | 1 | 8 | 0.421 | 0.029 | 0.212 | 0.030 |
| 22768 | LOMROG12-PC09 | 1.980 | 233 | 17 | 1 | 6 | 0.435 | 0.027 | 0.231 | 0.029 |
| 22769 | LOMROG12-PC09 | 2.160 | 276 | 13 | 1 | 8 | 0.475 | 0.053 | 0.278 | 0.040 |
| [b]22770 | LOMROG12-PC09 | 3.400 | 603 | 11 | 1 | 9 | 0.404 | 0.025 | 0.202 | 0.010 |
| 21406 | LOMROG12-TWC03 | 0.095 | 38 | 7 | 1 | 14 | 0.193 | 0.017 | 0.077 | 0.010 |
| 21407 | LOMROG12-TWC03 | 0.495 | 60 | 7 | 1 | 14 | 0.332 | 0.017 | 0.168 | 0.022 |
| | **Iceland Sea** | | | | | | | | | |
| 22745 | ODP 151/ 907A | 1.050 | 42 | 15 | 1 | 7 | 0.200 | 0.008 | 0.082 | 0.007 |
| 22746 | ODP 151/ 907A | 1.870 | 107 | 13 | 2 | 15 | 0.281 | 0.005 | 0.114 | 0.003 |
| 22747 | ODP 151/ 907A | 2.350 | 131 | 18 | 2 | 11 | 0.323 | 0.012 | 0.145 | 0.009 |
| 22748 | ODP 151/ 907A | 3.060 | 168 | 15 | 0 | 0 | 0.355 | 0.011 | 0.171 | 0.009 |
| 22749 | ODP 151/ 907A | 5.060 | 309 | 17 | 2 | 12 | 0.395 | 0.016 | 0.192 | 0.014 |
| 22750 | ODP 151/ 907A | 6.810 | 398 | 12 | 1 | 8 | 0.388 | 0.013 | 0.183 | 0.016 |
| 22751 | ODP 151/ 907A | 15.210 | 781 | 10 | 3 | 30 | 0.482 | 0.036 | 0.250 | 0.046 |
| | **Greenland Sea** | | | | | | | | | |
| 22732 | PS17 / 1906-2 | 0.150 | 10 | 12 | 5 | 42 | 0.130 | 0.005 | 0.065 | 0.002 |
| 22733 | PS17 / 1906-2 | 1.805 | 85 | 17 | 0 | 0 | 0.263 | 0.008 | 0.111 | 0.008 |
| 22734 | PS17 / 1906-2 | 2.005 | 111 | 12 | 0 | 0 | 0.306 | 0.006 | 0.138 | 0.005 |
| 22735 | PS17 / 1906-2 | 2.105 | 117 | 12 | 4 | 33 | 0.312 | 0.010 | 0.141 | 0.009 |
| 22736 | PS17 / 1906-2 | 2.205 | 122 | 12 | 0 | 0 | 0.331 | 0.032 | 0.156 | 0.028 |
| 22737 | PS17 / 1906-2 | 3.290 | 207 | 8 | 0 | 0 | 0.338 | 0.017 | 0.168 | 0.011 |
| 22738 | PS17 / 1906-2 | 3.605 | 225 | 12 | 2 | 17 | 0.355 | 0.010 | 0.178 | 0.010 |
| 22739 | PS17 / 1906-2 | 5.505 | 398 | 5 | 1 | 20 | 0.439 | 0.030 | 0.227 | 0.036 |
| | | | | | | | | | | |
| *Cibicidoides wuellerstorfi* | | | | | | | | | | |
| | **Alpha Ridge** | | | | | | | | | |
| 17341 | AO16-9-PC1 | 0.190 | 16 | 10 | 0 | 0 | 0.235 | 0.061 | 0.099 | 0.040 |





| | | | | | | | | | |
|---|---|---|---|---|---|---|---|---|---|
| 17342 | AO16-9-PC1 | 0.210 | 22 | 10 | 1 | 10 | 0.326 | 0.027 | 0.158 | 0.020 |
| 17343 | AO16-9-PC1 | 0.970 | 127 | 10 | 0 | 0 | 0.502 | 0.005 | 0.311 | 0.010 |
| 17344 | AO16-9-PC1 | 1.260 | 188 | 10 | 0 | 0 | 0.522 | 0.007 | 0.321 | 0.008 |
| 17346 | AO16-9-PC1 | 2.415 | 599 | 10 | 2 | 20 | 0.523 | 0.101 | 0.302 | 0.094 |
| | **Lomonosov Ridge** | | | | | | | | |
| 17331 | AO16-5-PC1 | 0.090 | 30 | 10 | 1 | 10 | 0.342 | 0.012 | 0.155 | 0.006 |
| 17333 | AO16-5-PC1 | 1.370 | 105 | 7 | 1 | 14 | 0.525 | 0.020 | 0.339 | 0.032 |
| 22754 | LOMROG12-PC03 | 0.520 | 66 | 13 | 3 | 23 | 0.412 | 0.014 | 0.208 | 0.013 |
| 17328 | LOMROG12-PC03 | 0.540 | 98 | 8 | 0 | 0 | 0.433 | 0.009 | 0.234 | 0.009 |
| 17329 | LOMROG12-PC03 | 0.741 | 198 | 8 | 1 | 13 | 0.485 | 0.008 | 0.284 | 0.008 |
| 17330 | LOMROG12-PC03 | 1.271 | 63 | 8 | 1 | 13 | 0.541 | 0.010 | 0.328 | 0.019 |
| [b]22755 | LOMROG12-PC03 | 1.441 | 247 | 9 | 1 | 11 | 0.491 | 0.017 | 0.267 | 0.035 |
| 22756 | LOMROG12-PC09 | 0.800 | 73 | 12 | 1 | 8 | 0.433 | 0.011 | 0.236 | 0.011 |
| 22757 | LOMROG12-PC09 | 1.250 | 140 | 22 | 2 | 9 | 0.476 | 0.007 | 0.266 | 0.011 |
| 22758 | LOMROG12-PC09 | 1.980 | 224 | 5 | 0 | 0 | 0.508 | 0.015 | 0.281 | 0.015 |
| 22752 | LOMROG12-TWC03 | 0.025 | 16 | 13 | 1 | 8 | 0.144 | 0.012 | 0.047 | 0.006 |
| 22753 | LOMROG12-TWC03 | 0.095 | 38 | 18 | 3 | 17 | 0.157 | 0.014 | 0.053 | 0.006 |
| 17327 | LOMROG12-TWC03 | 0.215 | 44 | 8 | 0 | 0 | 0.341 | 0.006 | 0.161 | 0.006 |
| | **Iceland Sea** | | | | | | | | |
| 22740 | ODP 151/ 907A | 1.050 | 42 | 9 | 3 | 33 | 0.166 | 0.026 | 0.059 | 0.013 |
| 22741 | ODP 151/ 907A | 1.870 | 107 | 21 | 1 | 5 | 0.337 | 0.015 | 0.151 | 0.010 |
| [b]22742 | ODP 151/ 907A | 2.350 | 131 | 15 | 6 | 40 | 0.160 | 0.013 | 0.064 | 0.006 |
| 22743 | ODP 151/ 907A | 3.060 | 168 | 10 | 1 | 10 | 0.388 | 0.025 | 0.198 | 0.026 |
| 22744 | ODP 151/ 907A | 15.210 | 781 | 18 | 1 | 6 | 0.549 | 0.016 | 0.345 | 0.032 |
| | **Greenland Sea** | | | | | | | | |
| 22724 | PS17 / 1906-2 | 0.150 | 10 | 15 | 0 | 0 | 0.175 | 0.009 | 0.059 | 0.004 |
| 22725 | PS17 / 1906-2 | 1.805 | 85 | 20 | 5 | 25 | 0.328 | 0.010 | 0.144 | 0.007 |
| 22726 | PS17 / 1906-2 | 2.005 | 111 | 20 | 1 | 5 | 0.351 | 0.016 | 0.160 | 0.007 |
| 22727 | PS17 / 1906-2 | 2.105 | 117 | 18 | 1 | 6 | 0.357 | 0.009 | 0.171 | 0.007 |
| 22728 | PS17 / 1906-2 | 2.205 | 122 | 21 | 1 | 5 | 0.379 | 0.016 | 0.191 | 0.015 |
| 22729 | PS17 / 1906-2 | 3.290 | 207 | 6 | 1 | 17 | 0.443 | 0.050 | 0.258 | 0.056 |
| [b]22730 | PS17 / 1906-2 | 3.605 | 225 | 6 | 0 | 0 | 0.343 | 0.014 | 0.160 | 0.023 |
| 22731 | PS17 / 1906-2 | 5.505 | 398 | 6 | 0 | 0 | 0.517 | 0.018 | 0.331 | 0.026 |

**Table 2: Extent of racemization (D/L) for aspartic acid (Asp) and glutamic (Glu) acid in samples of *Neogloboquadrina pachyderma* and *Cibicidoides wuellerstorfi* collected from the Arctic Ocean. Reported ages are from published age models.**

**[a] Number of subsamples used to calculate the mean and standard deviation**
**[b] Stratigraphically reversed sample**

b) Inter-species comparison

The overall subsample rejection rate was higher for *N. pachyderma* (18.2 %) than for *C. wuellerstorfi* (9.8 %) (Table 3). Only one C. *wuellerstorfi* subsample was rejected due to high serine content, significantly fewer than those for *N. pachyderma*, implying that secondary amino acids ('contamination') were not introduced during core storage or laboratory analysis because both species were treated similarly. *C. wuellerstorfi* subsamples were rejected more frequently as statistical outliers than were *N. pachyderma*. This could reflect the true heterogeneity

of ages of individual tests, which is revealed when fewer individuals are averaged together in a single subsample, especially where sedimentation rates are low (e.g. Dolman et al., 2021).




| Species | Total number of subsamples | L-Ser / L-Asp ≥ 0.8 | Non-covarying D/L Asp and D/L Glu | D/L Asp or D/L Glu not within ±2σ of sample mean | Subsample destroyed during analysis |
|---|---|---|---|---|---|
| *N. pachyderma* | 652 | 82 | 22 | 15 | 8 |
| *C. wuellerstorfi* | 376 | 1 | 18 | 18 | - |

**Table 3: Number of rejected subsamples per rejection criterion**

The proportion of samples with high intra-sample variability (coefficient of variation for D/L Asp and Glu > 10 %, following data screening) was approximately twice as high for *N. pachyderma* as for *C. wuellerstorfi*.

The rate of amino acid racemization varies between different foraminifera species (King and Neville, 1977). As some samples (n = 21) from certain stratigraphic intervals contained specimens of both *N. pachyderma* and *C.*
*wuellerstorfi*, this allowed a direct comparison of the rates of racemization in the two species (Fig. 3). One stratigraphically reversed sample was omitted prior to analysis. The slope of the least-squares regression fit to the D/L versus D/L data indicates that Asp racemized, on average, 16±2 % (n = 20) faster in *C. wuellerstorfi* than in *N. pachyderma*, and similarly, Glu also racemized faster (23±3 %) in *C. wuellerstorfi* than in *N. pachyderma*.

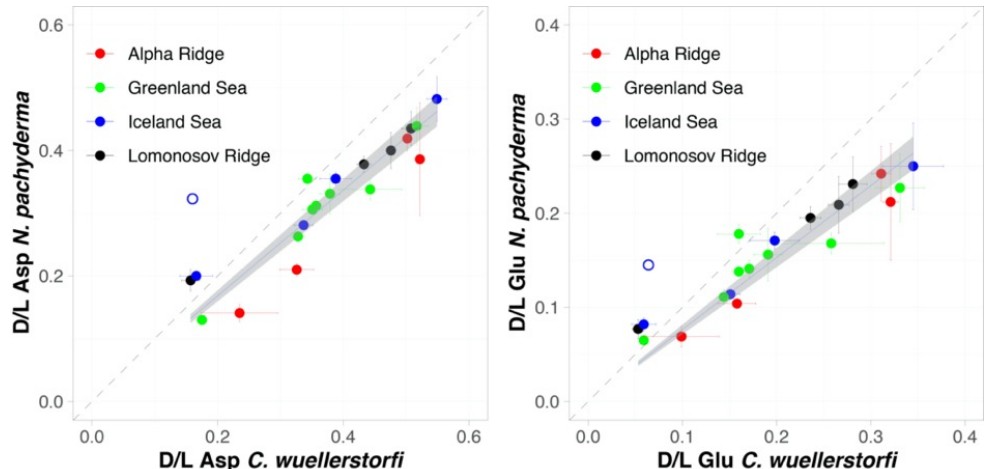


**Figure 3: Extent of racemization for aspartic acid (Asp) and glutamic acid (Glu) in *Neogloboquadrina pachyderma* and *Cibicidoides wuellerstorfi* from samples from the same stratigraphic depths collected from different regions of the Arctic Ocean. Error bars represent ±1σ intra-sample variability, and the dashed line corresponds to the line of equality. Open symbols mark the sample excluded from regression analysis.**


c)    Relationship between D/L values and age / stratigraphic depth

The extent of racemization (D/L values) for Asp and Glu show a systematic increasing trend with increasing sample age in both species from the Greenland and Iceland seas, and follow a simple power function (Fig. 4).


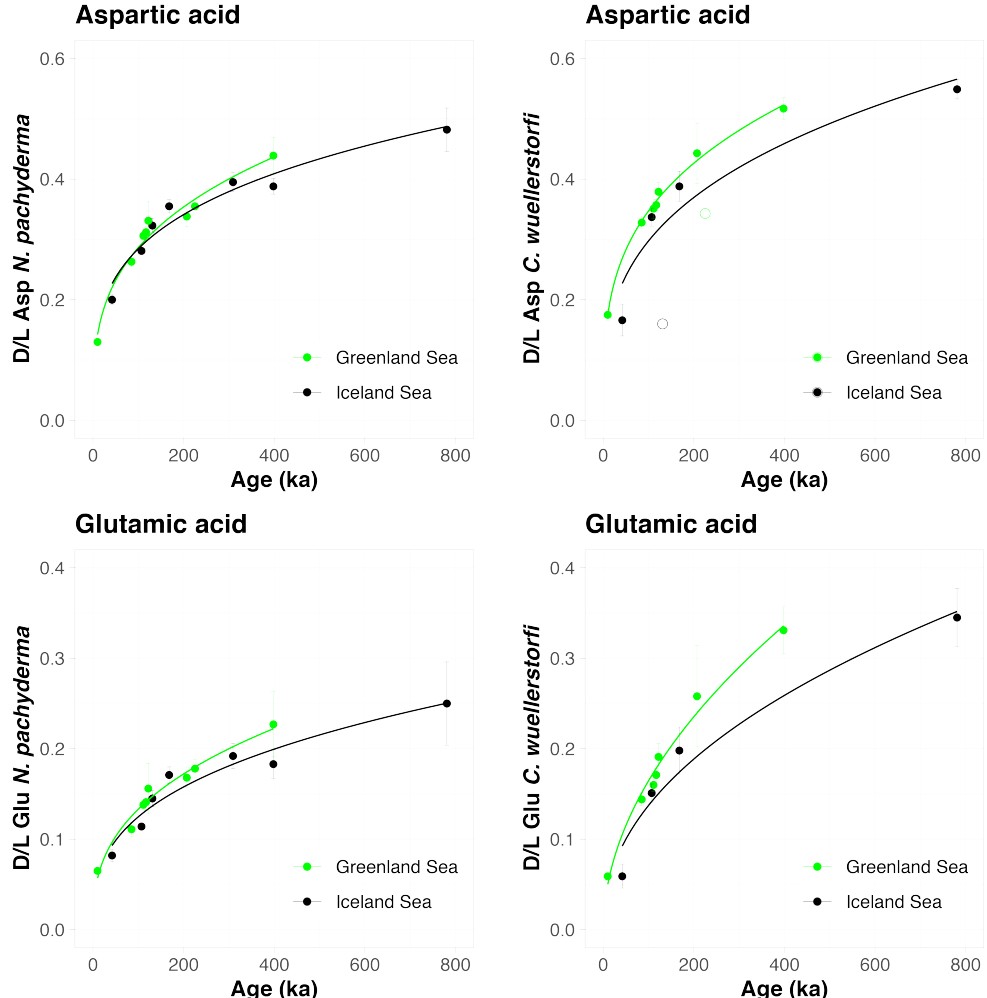

**Figure 4: Extent of racemization for aspartic acid (Asp) and glutamic acid (Glu) in samples of *N. pachyderma* and *C. wuellerstorfi* from sediment cores PS17/1906-2 (Greenland Sea) and ODP 151/907A (Iceland Sea). Error bars represent ±1σ intra-sample variability. Stratigraphically reversed samples are marked with unfilled symbols and were excluded to maximise the goodness of fit.**


D/L values obtained from sediment cores from the central Arctic Ocean, where age-depth models are less certain, are displayed on the ACEX composite depth scale based on correlations using bulk density profiles (Fig. 5). The extent of racemization follows a simple power function (stratigraphically reversed samples excluded), as also

observed in samples from the Nordic Seas, and D/L values generally overlap between correlative samples from multiple cores. This directly supports the accuracy of the lithostratigraphic correlation established between the sites. D/L values from *N. pachyderma* appear to reach a plateau at ~0.4 for Asp and ~0.2 for Glu (~6 m depth in the ACEX record). Plateauing of D/L Asp values was previously documented in other Arctic Ocean cores as well (Kaufman et al., 2008). It is unclear whether plateauing is present in *C. wuellerstorfi* samples, as only a few





samples were analysed in the corresponding age range, and the oldest sample has the largest intra-sample variability.

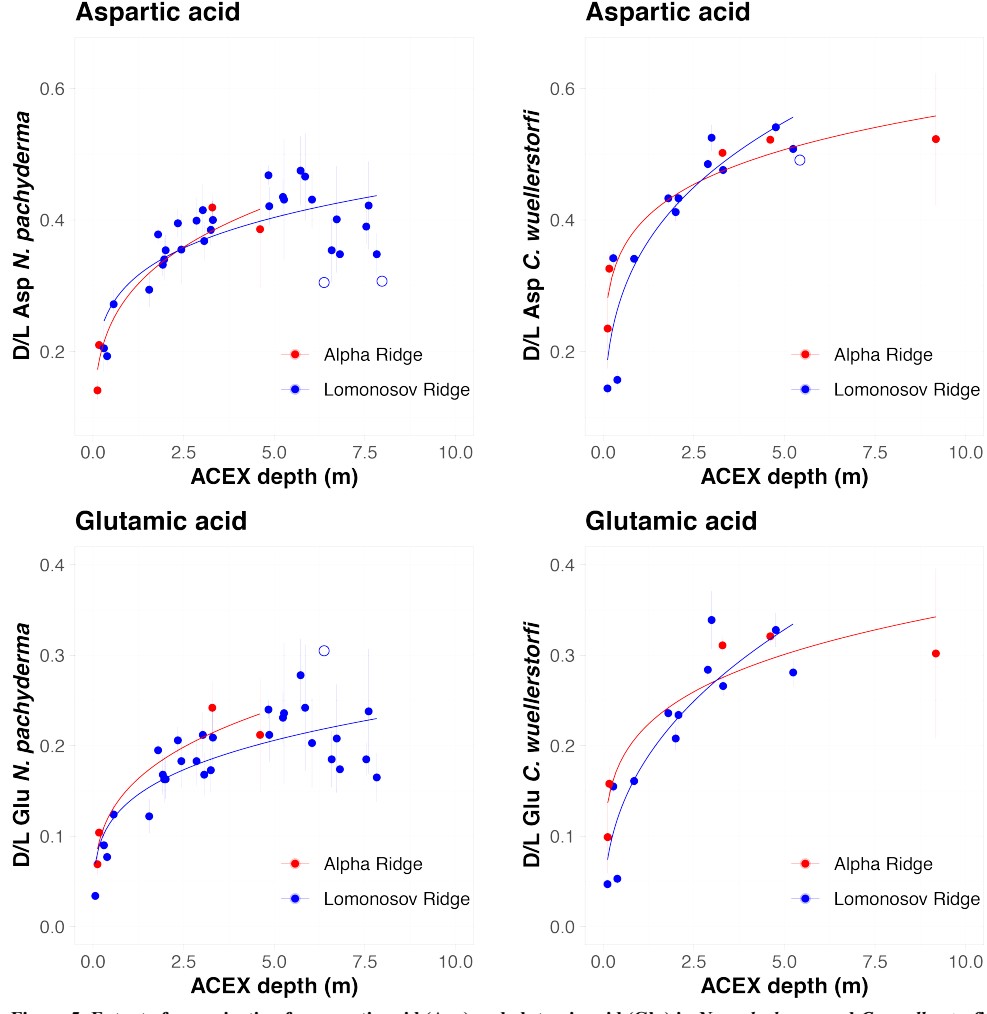

**Figure 5: Extent of racemization for aspartic acid (Asp) and glutamic acid (Glu) in *N. pachyderma* and *C. wuellerstorfi***
**from the central Arctic Ocean displayed on the ACEX composite depth. Sediment cores from investigated in this study were grouped based on their geographical location (Alpha Ridge – red, Lomonosov Ridge – blue). Error bars are ± 1σ intra-sample variability, and unfilled symbols mark stratigraphically reversed samples.**

### 4. Discussion


Aspartic acid and glutamic acid racemization analyses of multiple foraminifera taxa obtained from cold bottom water sites across the globe showed that sample ages can be confidently estimated by calibrated age equations, which relate the extent of racemization of the amino acids to independently determined sample age (Kaufman et al., 2013). These globally derived age equations are based on both planktic and benthic foraminifera species with





*N. pachyderma* contributing up to ~21% of all samples, but exclude *C. wuellerstorfi*. Ages derived using the global
       age equations generally agree with independently derived ages at the Yermak Plateau (West et al., 2019), but do
       not agree with working age models applied to the central Arctic Ocean, beyond about 40 ka. If existing age models
       in the central Arctic Ocean are correct, the extent of racemization for *N. pachyderma* is higher than expected when
       compared to other oceans (Kaufman et al., 2008, 2013).


       The results of AAR presented here show that Asp and Glu racemize in a predictable manner in both *N. pachyderma*
       and *C. wuellerstorfi* samples from the Nordic Seas (Fig. 4), although the rates differ between the species (Fig. 3).
       Both D/L Asp and D/L Glu values obtained from *N. pachyderma* samples from the Nordic Seas clearly follow a
       trend previously observed at the Yermak Plateau (Fig. 6), implying that racemization kinetics are indistinguishable

in these areas. These data from independently dated cores from cold-water sites provide further support for the
       integrity of the AAR technique, and for the globally calibrated age equations of Kaufman et al. (2013).

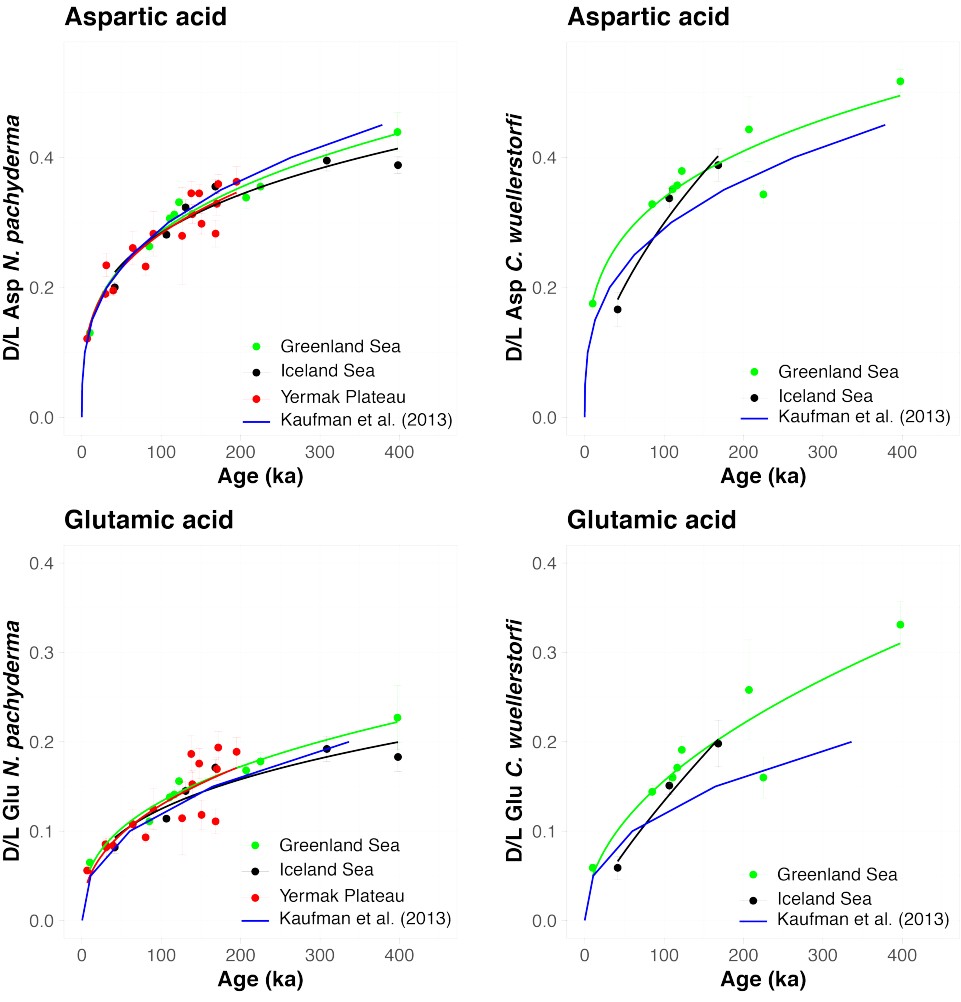





**Figure 6: Extent of racemization in aspartic acid (Asp) and glutamic acid (Glu) in *Neogloboquadrina pachyderma* and**
***Cibicidoides wuellerstorfi* from the Greenland and Iceland seas and the Yermak Plateau. D/L Asp and Glu values and**
**associated sample ages, as predicted by the globally calibrated age equations of Kaufman et al. (2013), are shown in**
**blue. Data for the Yermak Plateau from West et al. (2019). Error bars represent ± 1σ intra-sample variability.**

The inter-species comparisons show that Asp racemizes approximately 16 % faster in *C. wuellerstorfi* than in *N.*
*pachyderma* (Fig. 3). Differences of similar magnitude were previously observed between *N. pachyderma* and
*Pulleniatina obliquiloculata*, in which Asp racemizes 12-16 % faster than in *N. pachyderma* (Kaufman et al.,
2013). Quantifying the relative differences of the racemization rates between the species can facilitate future AAR
studies. When a taxon is unavailable at a certain stratigraphic depth, the extent of racemization in one species
could be adjusted to that of another based on the difference in the observed rate of racemization. *C. wuellerstorfi*
was not included in the globally calibrated age equations and the higher rate of AAR in this species suggests that
D/L values should not be used in combination with the equations without adjustment (Fig. 6).

The purportedly higher rates of racemization in *N. pachyderma* from the central Arctic Ocean were argued to be
caused by factors other than taxonomic effects (Kaufman et al., 2013). Either racemization generally progresses
at a higher rate here due to an unknown reason, or existing age models used to constrain the rate of AAR from the
central Arctic Ocean underestimate the true age of deposition. Given the existing age models, higher-than-
expected rates of racemization are also observed in this study in both *N. pachyderma* and *C. wuellerstorfi* samples
from the central Arctic Ocean (Fig. 7). Differences between ages predicted by the global calibrated age equation
and currently available sample ages based on the ACEX age model are larger for *C. wuellerstorfi*, as expected,
since we established that AAR is faster in this taxon than in *N. pachyderma*.



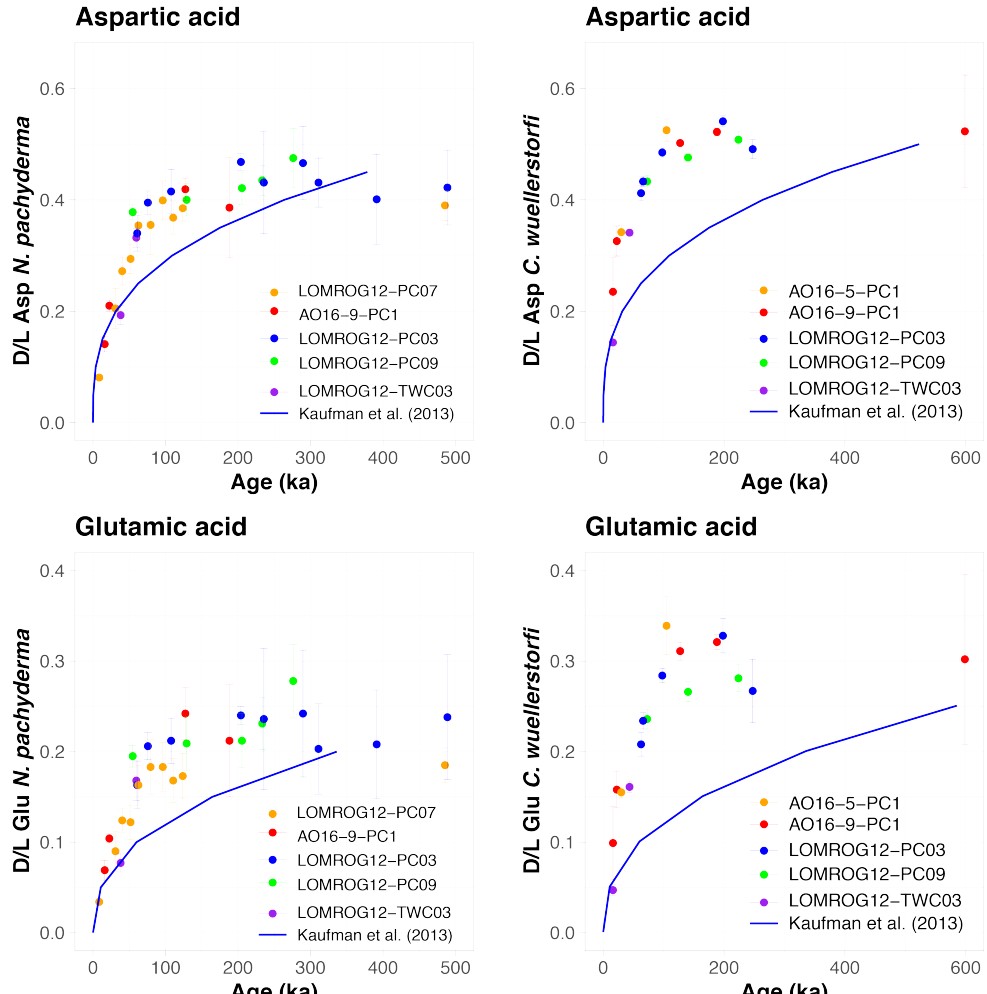

**Figure 7: Extent of racemization for aspartic acid (Asp) and glutamic acid (Glu) in *N. pachyderma* and *C. wuellerstorfi***
**from the central Arctic. D/L values are displayed against the ACEX age model. D/L values versus age trend defined by**
**the globally calibrated age equations (Kaufman et al., 2013) is shown in blue. Error bars represent ± 1σ intra-sample**
**variability.**

The AAR trends in *C. wuellerstorfi* samples from the Nordic Seas, where age models are better constrained, can
also be compared with those in the central Arctic Ocean samples. The racemization of Asp and Glu in *C.
wuellerstorfi* samples from both the Greenland and Iceland seas (GIS) follow the same trend (Fig. 4), which can
be approximated with regression analysis. Following the removal of stratigraphically reversed samples and
samples with high analytical uncertainty (n=5), simple power models (Eq. 1 and 2)


$$t = 3827.6 * (D/L\ Asp)^{3.395} \quad (1)$$
$$t = 4979.1 * (D/L\ Glu)^{2.138} \quad (2)$$



(where t = age in ka) fit the D/L Asp and D/L Glu versus age relationship well for the past 400 ka (the period for which these models are robust), and can be applied to D/L values in *C. wuellerstorfi* from the central Arctic Ocean. This reveals that sample ages are younger than predicted by the model based on the Nordic Sea samples (Fig. 8). The higher-than-expected D/L values observed for *C. wuellerstorfi* from the central Arctic Ocean confirm earlier findings, which argued that higher racemization rates in the central Arctic Ocean are not the result of taxonomic effects (Kaufman et al., 2013).

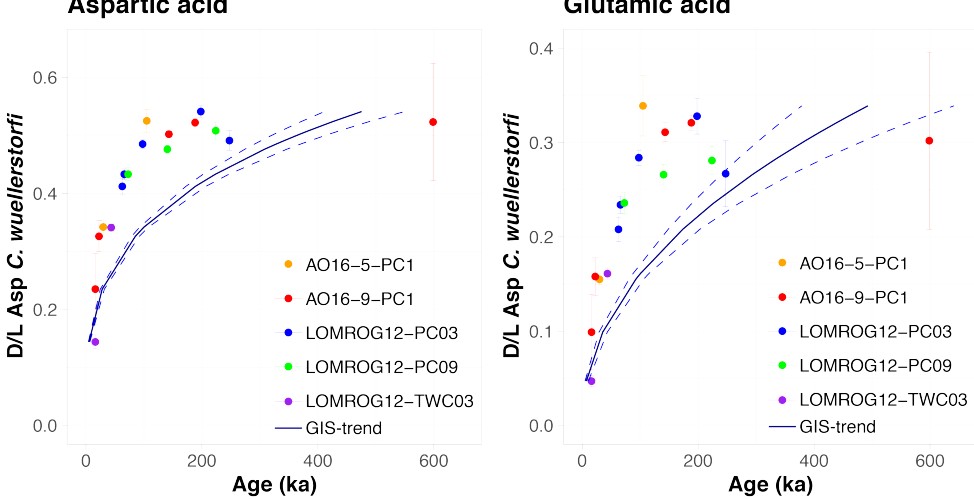

**Figure 8. Aspartic acid (Asp) and glutamic acid (Glu) racemization versus sample age in *C. wuellerstorfi* from the central Arctic Ocean. Model predictions based on the trend defined by samples from the Greenland and Iceland seas is also shown ('GIS trend' – blue line; blue dashed lines mark 95% confidence intervals). Error bars represent ± 1σ intra-sample variability.**

The discrepancy between the globally (*N. pachyderma*) and GIS (*C. wuellerstorfi*) calibrated AAR ages and the established chronology for central Arctic Ocean sediments is illustrated in Figure 9 and Table 4. The calibrated AAR ages suggest that what is interpreted as substages in MIS 5 are instead separate interglacial periods. This interpretation would indicate that the diamict unit previously assigned to MIS 6, and representing the onset of a fundamentally different depositional regime in the Arctic characterised by recurrent coarse-grained facies (O'Regan et al., 2010), is likely older than MIS 8 and possibly as old as MIS 12. Exact age determinations are difficult due to a paucity of data from these lower depths, and increased downcore scatter in the stacked AAR results (*discussed below*).

It cannot be excluded that the established chronologies of the central Arctic Ocean sediments underestimate their true ages, but significant shifts (representing multiple glacial cycles) in sediment ages are required and result in a number of fundamental inconsistencies when compared to other geochronological data. For example, recent work (O'Regan et al., 2020) placed the first occurrence of the coccolithophore, *Emiliania huxleyi*, in core LOMROG12-PC03 at 1.39±0.02 mbsf in marine isotope stage (MIS) 7. The AAR age equations (1) and (2), defined by the trend in *C. wuellerstorfi* samples from the Greenland and Iceland seas, yield estimated ages of 475±12 ka (D/L Asp) or



459±21 ka (D/L Glu) (MIS 12/13) based on the mean D/L ±1SE for a sample from 1.27 m in this core. Similarly, the globally calibrated age equation (Kaufman et al., 2013) returns an MIS 12 age of 427±20 ka for *N. pachyderma* samples from 1.29 m in LOMROG12-PC03. These ages are substantially older than the global evolutionary

occurrence of *E. huxleyi* during MIS 8 (Thierstein et al., 1977; Anthonissen and Ogg, 2012). Furthermore, *E. huxleyi* is abundant in the 0.58 – 0.82 mbsf core interval of LOMROG12-PC03, which was assigned an MIS 5 age (O'Regan et al., 2020). In contrast, the AAR trend from the Greenland and Iceland seas assigns ages of 189±6 – 328±7 ka (MIS 6 – 9) for the 0.52 – 0.74 m core depths (Table 4). Although this could be reconciled with the existence of *E. huxleyi* in this interval, assuming it entered the Arctic shortly after its first evolutionary occurrence,

the MIS 6-9 age is not consistent with results from optically stimulated luminescence dating in stratigraphically co-eval sediments from another core on the Lomonosov Ridge which support the MIS 5 age assignment (Jakobsson et al., 2003).

| Stratigraphic depth (m) | ACEX equivalent depth (m) | Estimated age (ka) | | |
|---|---|---|---|---|
| | | ACEX age model (Backman et al., 2008; O'Regan et al., 2008) and *E. huxleyii* occurrences (O'Regan et al., 2020) | AAR geochronology – aspartic acid in *N. pachyderma* (Kaufman et al., 2013) | AAR geochronology – aspartic acid in *C. wuellerstorfi* (this study) |
| 0.52-0.74 | 1.96-3.35 | MIS 5 | MIS 6-9 | MIS 6-9 |
| 1.27-1.39 | 4.62-5.44 | MIS 7 | MIS 9-12 | MIS 9-12 |

**Table 4: Age estimates for selected stratigraphic intervals in LOMROG12-PC03 based on the ACEX age model and occurrences of *E. huxleyi*, and AAR geochronology. MIS = marine oxygen isotope stage.**

On the other hand, sedimentation rates predicted by the AAR global and GIS calibration equations are consistent with radiocarbon derived estimates over the past 40 ka, but beyond this point remain intermediate between the proposed age model for the ACEX record (O'Regan et al., 2008) and those recently inferred from $^{230}$Th and $^{231}$Pa

extinction ages in sedimentary sequences from this region of the Lomonosov Ridge (Hillaire-Marcel et al., 2017; Purcell et al., 2022). Significant work is required to understand the origin and resolve the differences between these geochronological approaches. However, a critical observation is that even the older estimated ages from the global and GIS calibration equations remain considerably younger than the long-overturned paleomagnetic-derived age model that would place the Bruhnes-Matuyama boundary (~780 ka) at approximately 5 m depth in

the ACEX record (O'Regan et al., 2008).

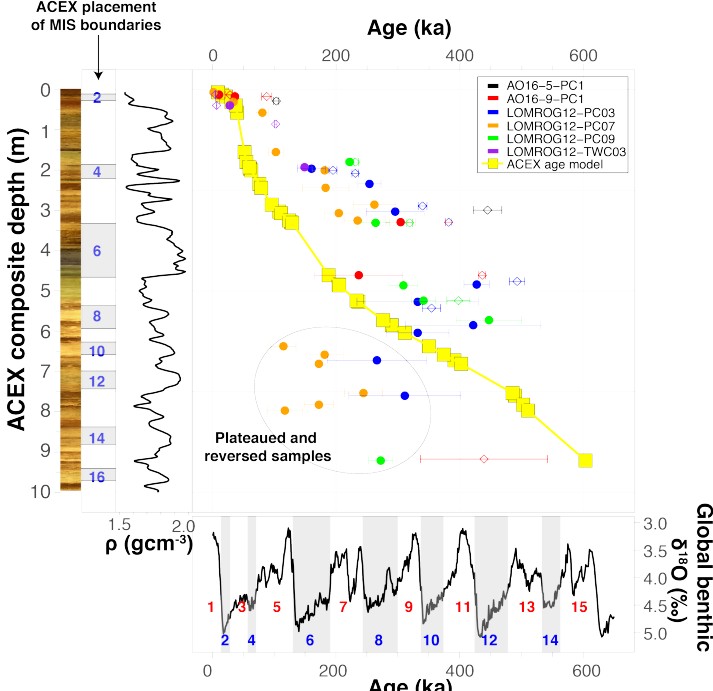

**Figure 9: Alternative age – depth relationships in sediment cores from the central Arctic Ocean. Circles and diamonds mark ages estimated by globally calibrated rates of racemization of aspartic acid in *N. pachyderma* (Kaufman et al., 2013) and GIS-calibrated rates of racemization of aspartic acid in *C. wuellerstorfi* (this study) respectively. Error bars represent 95 % confidence intervals. The ACEX age model is based on Backman et al., 2008, Frank et al., 2008, and O'Regan et al., 2008. Also shown are ACEX digital core image, marine oxygen isotope stage (MIS) boundaries based on the ACEX age model, bulk density (ρ) profile of the ACEX core, and the global benthic δ$^{18}$O record and corresponding oxygen isotope stages (MIS 1-15) based on Lisiecki & Raymo (2005).**

An alternative to central Arctic Ocean age models underestimating the true age of sediments is that the assumption that the rate of AAR in foraminifera from the central Arctic Ocean is the same as from other globally distributed cold water sites (Kaufman et al., 2013) could be invalid. A fundamental premise of this study is that the investigated deep-sea sites have experienced similar temperatures, which changed similarly over time. Modern bottom water temperatures near the coring sites are very similar (Table 1), but they may have differed in the past. Cronin et al. (2012) showed that the central Arctic basin waters, at ∼1000-2500 m depth interval, were 1-2°C warmer during the past 50 – 11 ka than today. While this might account for part of the apparently higher rate of AAR in the central Arctic Ocean, West et al. (2019) showed that more substantial differences (∼ >4 °C) in effective diagenetic temperatures would be required to achieve the differences observed between D/L values of equivalent age samples from sediment cores from the central Arctic Ocean and those from the Nordic Seas. Available heat flow data (Shephard et al., 2018) show that geothermal flux is not unusually high in the central Arctic Ocean when compared to the Yermak Plateau, and thus cannot explain the apparently higher rates of racemization. Therefore, while offsets in D/L values in stratigraphically co-eval sections of different central Arctic Ocean cores in this study might, in part, be attributed to differences in bottom water temperatures, such differences are unlikely able to explain the overall higher inferred rates of racemization compared to other global sites.



D/L values in many samples of *N. pachyderma* from the central Arctic Ocean cores exhibit a distinct plateauing below the ~5.5 – 6 m composite ACEX depth (Fig. 5 and 9). Some of these samples can be considered as stratigraphically reversed (i.e. their D/L values are lower than of those from shallower depths). Such plateauing

of D/L values was previously observed in other areas of the Arctic Ocean (Kaufman et al., 2008), but its cause remains unclear. While sediment mass movements, glacial erosion and subsequent re-deposition are known to occur across the Arctic Ocean (e.g. Jakobsson & O'Regan, 2016; Boggild et al., 2020; Pérez et al., 2020; Schlager et al., 2021), this explanation would require that almost all of the material above 5.5 – 6 m composite depth was reworked similarly across multiple cores, which seems untenable.


Sedimentation rates in the central Arctic Ocean are much lower than other, more marginal areas (Backman et al., 2004), and can be greatly reduced during glacials in some regions of the Arctic Ocean (Jakobsson et al., 2014). Conversely, thick diamict units suggest rapid influxes of ice rafted material during some glacials or glacial terminations. These punctuated episodes of sedimentation can not only introduce hiatuses, but also impact the

length of time biocarbonates are exposed to microbial activity on the seafloor (Sejrup and Haugen, 1994), which in turn can speed up or slow down organic diagenetic processes that influence the rate of AAR.  Local differences in sedimentation rates (and temperatures) might also account for some of the observed scatter in AAR results.

Microbial influences in central Arctic Ocean sediments might account for a unique control on racemisation rates

and could explain the large offsets in the inferred AAR rates between the Arctic and global oceans. More specifically, where sedimentation rates are low, greater microbial activity could increase organic diagenesis and lead to higher apparent rates of AAR. On the other hand, continuous turnover and reworking of microbial necromass within the foraminifera test could instead lower the apparent rate of racemization via regeneration of L amino acids, and alter the apparent AAR rates, as has been documented for bulk organic matter in marine

sediments (Braun et al., 2017). Furthermore, Kubota et al. (2016) isolated alphaproteobacteria from deep-sea sediments from the Sagami Bay (Japan), which exclusively utilised D amino acids as a carbon and nitrogen source. If such microbes were also present within foraminifera tests of the central Arctic Ocean, they could potentially alter the progress of racemization. Apparent plateauing of D/L values (e.g. Fig. 5, 7, 9) could be associated with these processes, but would require a distinctively different microbial sedimentary environment in the central

Arctic Ocean than elsewhere (such as the Yermak Plateau or Norwegian-Greenland Sea, where the global age-equation appears applicable). This is not improbable; for example, recently Yu et al. (2020) suggested that different marine environments could be characterised by distinct bacterial groups which utilise D-amino acids. Understanding of microbial communities and their interactions with their immediate environment in the Arctic Ocean is insufficient and requires future research.


If the rate of AAR in foraminifera is indeed higher in the central Arctic Ocean, its exact cause remains unknown, but the data reported in this study confirm that it is not caused by taxonomic effects, as it can be observed not only in the planktic *N. pachyderma* but also in the benthic species *C. wuellerstorfi*. We have highlighted the discrepancies that arise in central Arctic Ocean age models if the global (*N. pachyderma*) and GIS (*C.*

*wuellerstorfi*) AAR age equations are applied. The results warrant a critical evaluation of existing Arctic Ocean



age models and the need to more fully assess the environmental factors that may influence racemisation rates in central Arctic Ocean sediments.

### 5. Conclusions


- Aspartic and glutamic acids racemize faster in *C. wuellerstorfi* than in *N. pachyderma*, and the extent of racemization for these amino acids increases progressively with sample age in both species from multiple sediment cores. Their trends conform to a simple power function.

- *C. wuellerstorfi* samples are characterised by lower intra-sample variability than those of *N. pachyderma*, and this, coupled with a reduced subsample rejection rate and faster sample processing offered by its larger tests make it a preferred target of AAR studies.

- Ages of *N. pachyderma* samples from the Greenland and Iceland seas agree very well with ages predicted
by previous globally calibrated age equations for aspartic acid and glutamic acids (Kaufman et al., 2013), and confirm their applicability in these regions.

- The rate of racemization for *C. wuellerstorfi* was calibrated for the past 400 ka using samples from the Greenland and Iceland seas, where sample ages are robust, and D/L values conform tightly to a power
trend. Applying this calibration to the *C. wuellerstorfi* samples from the central Arctic indicates that they are older than their currently accepted ages. This confirms that higher-than-expected D/L values in the central Arctic Ocean are not the result of taxonomic effects.

**Acknowledgements**


Funding for this research was provided by the Swedish Research Council (grant number: DNR-2016-05092), the Bolin Centre for Climate Research (Ref. RA6_21_05), and the US National Science Foundation (1855381). We thank all expedition crew members and scientific parties who facilitated data collection. We further thank Jutta Wollenburg, Jens Matthiessen and the IODP Bremen Core Repository for providing samples from the Greenland
and Iceland seas, and Jordon Bright, Joshua Smith and Katherine Whitacre for laboratory assistance.

**Supplementary Material**

Supplementary material associated with this article can be found in the online version.


**Data availability**

The results of amino acid analyses of all 1028 subsamples included in this study will be archived at the World Data Service for Paleoclimatology (https://www.ncdc.noaa).




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
