# Peer review of "Amino acid racemization in *Neogloboquadrina pachyderma* and *Cibicidoides wuellerstorfi* from the Arctic Ocean and its implications for age models"

_Geochronology, 2022_

## Referee Comment (RC3)

Review of "Amino acid racemization in Neogloboquadrina pachyderma
and Cibicidoides wuellerstorfi from the Arctic Ocean and its
implications for age models» by West et al. for Geochronology

This paper present new amino acid data from a series of sediment cores from the Arctic
Ocean and the Nordic Seas. The new data include analyses of the planktonic species N.
pachyderma and the benthic species C. wuellerstorfi and it is focused on the racemization
reaction of Aspartic and Glutamic acid. The results are discussed in the context of the
challenges in establishing robust chronologies for Arctic Ocean deep sea sediments. The
paper is well written and has a sound scientific approach and should definitely be published.
However, after reading the paper I think the authors should be a little bit stronger on their
conclusion regarding the implications of their findings. To me it seems clear that the results
suggest that either the established chronology is to young or the bottom waters have been
warmer.

In the revision of the paper I would like the authors to consider expand with the following:
1. It is fine that the aa data generally support the correlation of cores based on density.
   However, merging all data  on the depth scale of the ACEX core may introduce more
   scatter and uncertainty than needed.
2. Present all Arctic Ocean cores with the basis for correlation also on depth scales.
3. Expand the section with some hard data on the basis for the ACEX established
   chronology and expand the discussion of the aa results relative to the other methods
   used.
4. A review/figure/profile depicting the present main water masses in the Arctic Ocean
   may be useful. Perhaps some references to modelling work concerning possible
   temperature conditions in the Arctic ocean during the glacial stages may be useful-

Miinor suggestions on text

30      … bottom water temperatures *may have been* similar.

33      ….models. *Also* a better understanding of temperature histories at the investigated
sites and other possible environmental factors that may influence rasemisation rates in the
central Arctic ocean, is needed.

64      Bottom water of Atlantic and Pacific sites are presently generally a few degrees
warmer than the Nordic Seas and Arctic ocean.

100     …..influence of *Atlantic surface water*, and…

110      …cores *have been* developed….

141     ..with *21(?) samples*…

172     Where the reversal confirmed for both species at the same level?

214     Considering what we know about past and present hydrography the samples from the Nordic Seas most likely have been exposed to the same water masses through time. Also the datapoints you have, seems not allow for establishing different pathways at the two sites. Suggest you make one polynomial fit but keep the coloring of points.

249     This is surprising taken into account that the temperatures potentially have been lower than the "global" ocean.

Fig. 2
Suggest that the density data is plotted on individual core depths with correlation to ACEX indicated with lines.

Fig. 5
It would have been nice to see the data for each core plotted on core depth in a separate figure..

---

## Author Comment (AC1)

We appreciate the time the reviewer took to review the manuscript and thank the reviewer for their constructive comments. We have tried to address them all and feel that they have improved the manuscript.

The reviewer's comments are in black italics, followed by our responses in blue. A revised manuscript with tracked changes is also included.

*Anonymous Referee 1*

*The paper presents new geochronological data in the form of amino acid racemisation of N. pachyderma and C. wuellerstorfi from a range of cores in the Arctic Ocean. New age estimates are derived for these cores and the discrepancies between different dating techniques in this area are addressed in detail. This work also demonstrates the utility of C. wuellerstorfi for AAR, a species which has not been previously investigated in detail using modern separation methods. I therefore recommend this paper for publication with minor revisions, detailed below.*

*Specific comments:*

*Line 159: It should be stated that during hydrolysis, Asn and Gln irreversibly hydrolyse to Asp and Gln, so the 'Asp' and 'Glu' reported in this study also includes any Asn and Gln present in the biomineral (see Hill 1965 https://doi.org/10.1016/S0065-3233(08)60388-5).*

> We altered to text to include that 'Asp' and 'Glu' reported in this study also include any Asn and Gln present in the biomineral.

*Table 3: Table 3 gives a count of subsamples destroyed during analysis, but this is not mentioned in the text.*

> We have omitted any mention of destroyed subsamples as these do not influence the data.

*Figures 4 and 5: The authors mention the reduced confidence in the power function for C. wuellerstorfi from the Alpha Ridge due to the small number of C. wuellerstorfi samples analysed at this site; as only 4-5 samples of N. pachyderma were analysed from the Alpha Ridge and of C. wuellerstorfi from the Iceland Sea, this uncertainty should also be discussed for these sites/species.*

> The power model for central Arctic Ocean cores was used in combination with sample depth rather than proposed sample age due to the reduced confidence in the ages. We do not specifically mention reduced confidence in the power function for *C. wuellerstorfi* due to small number of samples analysed from this area.

*Lines 293-300: Equations for a simple power model applied to the data are given here. These should be given when the models are first introduced (in Section 3c). I would also suggest briefly summarising the precedent of using simple power functions to model racemisation (e.g. Clarke and Murray-Wallace 2006 https://doi.org/10.1016/j.quageo.2006.12.002), justifying why the authors used this model and how the exponent for each model was derived.*

> In Section 3c we show that the trends at individual areas conform to simple power functions, characteristic of racemization under isothermal conditions (e.g. Kaufman, 2006; Clarke and Murray-Wallace, 2006). We have added these references to the text. The power model used in the Discussion (equations 1 and 2) combines samples from the Greenland and Iceland seas, and only applies to samples up to 400 ka old. Hence, we prefer to keep these in the Discussion. Exponents were obtained using regression, as stated in the text.

***Line 391-409:*** *The authors suggest that differences in microbial environments during diagenesis may account for some of the discrepancy between racemisation and other dating methods. As this should only affect open-system inter-crystalline material, it would be worth addressing recent work on isolating the intra-crystalline fraction in foraminifera (see Penkman et al. 2008 https://doi.org/10.1016/j.quageo.2007.07.001 for IcPD overview and Wheeler et al. 2021 https://doi.org/10.1016/j.quageo.2020.101131 for IcPD of* N. pachyderma*), as the intra-crystalline approach may minimise or eliminate environment-specific effects on racemisation rates.*

> **We extended the Discussion to include that isolation of the intra-crystalline fraction, which behaves as a closed system during diagenesis (Penkman et al., 2008; Wheeler et al., 2021), by means of bleaching could minimise the influence of bacterial activity on racemization rates. However, recent work (Millman et al., 2022) showed that bleaching does not necessarily improve the quality of the results, thus the current study used the standard weak oxidative pre-treatment.**

***Technical corrections:***

***All figures:*** *increase the line thickness of error bars – they are very difficult to see, especially for green/yellow data. As figure 8 presents data from only one species (thus placing Asp on the lefthand plot and Glu on the righthand plot), I would recommend switching the layout of the other plots that the amino acid is faceted horizontally and the species vertically, so that all figures are consistent in this respect.*

> **We have increased the line thickness of the error bars on all figures, and rearranged the figures as suggested by the reviewer.**

***Line 19:*** *'large geographical area, from the Greenland'.*

> **Corrected.**

***Lines 23, 241, 244, 403:*** *'foraminifer species/taxa/tests' or 'species/taxa/tests of foraminifera' - the singular 'foraminifer' should be used in the adjectival form.*

> **We prefer to use the form 'foraminifera' for consistency. We refer to Lipps et al. (2011 DOI:10.2113/gsjfr.41.4.309).**

***Line 41-42:*** *'the protein amino acid isoleucine over time in samples of the planktic foraminifer* Neogloboquadrina pachyderma *and the benthic species* Cibicidoides wuellerstorfi'.

> **Corrected.**

***Line 69:*** *Consider removing the dashes around 'undetermined' to improve the readability of this sentence.*

> **Removed.**

***Line 110-111:*** *Consider listing the Nordic Sea cores here explicitly, e.g. 'Cores from the Nordic Seas (ODP151/907A and PS17/1906-2) primarily relied on oxygen isotope stratigraphy'.*

> **Changed.**

***Line 122, and in general:*** *Consider putting references at the end of a sentence/clause to improve readability.*

> **Changed.**

***Line 149, and in general:*** *L should be capitalised in mL and μL.*

> **Changed.**

***Line 155-159:*** *Consider breaking the sentence starting 'The peak-area ratio…' into two sentences at 'extent of racemisation, but this study'.*

> **Changed.**

***Table 3:*** *Full stop at end of table caption.*

> **Corrected.**

***Line 200:*** *Consider giving the name of the stratigraphically reversed sample removed from the species comparison.*

> **Changed.**

***Figures 4, 5 and 9:*** *Increase line thickness on open circles/diamonds – these are challenging to see, especially for the pale data points.*

> **Corrected.**

***Figure 6:*** *Define black/green/red lines in each figure. Also consider changing line style of the blue line (e.g. to dashed or dotted) to make it clear that it is an age model derived from other data, especially as the colour of this line is to denote data from this paper in other figures.*

> **We included in the figure caption that the green, black and red lines mark power function fits for the various areas.**

***Line 264:*** *In abstract, a standard deviation is given for the difference between racemisation rates of the two species; this should be quoted here rather than 'approximately 16 %'.*

> **The SD is already stated in the Results and here we are simplifying for purposes of the Discussion.**

***Line 280:*** *Consider changing 'since we established that AAR is faster' to 'as racemisation proceeds more quickly'.*

> **Changed to '*racemization proceeds faster in this taxon than…*'**

***Figures 7 and 8:*** *Consider adjusting palette – blue and purple are very similar, and yellow and green challenging to separate under red-green colourblind conditions. I would also recommend not reusing blue for the LMROG12-PC03 data and the Kaufman 2013 model, as this implies a connection between them.*

> **We changed the colour palette to a colour-blind-friendly one.**

***Line 315:*** *'what are interpreted as'*

**We altered the sentence to refer to '*intervals interpreted as substages in MIS 5*'.**

***Line 325:*** *Make sure that 'marine isotope stage' is defined in its first instance and that the acronym is used thereafter.*

**Defined in Line 66.**

***Table 4:*** *Left-align text on row 2, as justified text is difficult to read in narrow columns.*

**Changed.**

***Line 384:*** *Consider changing 'seems untenable' to 'is unlikely' or 'is highly unlikely' – more scientific language.*

**Changed to "*highly unlikely*".**

---

## Author Comment (AC2)

We appreciate the time Colin V. Murray-Wallace took to review the manuscript and thank the reviewer for his constructive comments. We have tried to address them all and feel that they have improved the manuscript.

The reviewer's comments are in black italics, followed by our responses in blue. A revised manuscript with tracked changes is also included.

*Colin V. Murray-Wallace*

*This is a very interesting and in many respects, throught-provoking manuscript because it tries to come to terms with a complex marine geological problem, while at the same time, resolve some of the typically challenging issues in Quaternary aminiostratigraphy. The latter involves the potentially different diagenetic temperatures and a genus-effect on racemization with their ultimate influence on the measured extent of amino acid racemization in fossils, and ultimatelty an assessment of age of the marine successions.*

*The manuscript discusses the potential difficulties of inferring the age of the successions in question and clearly outlines many geological and environmental attributes that confound a conclusive age interpretation. While concluding that the higher amino acid D/L values for the foraminifer C. wuellerstorfi is not due to a genus effect, a whole set of new questions arise to reconcile the basis for the extent of racemization observed in these individuals compared with other Arctic Ocean deep sea cores. In this sense, the existing manuscript is to some extent open-ended in its conclusions. Perhaps the conclusions can be more decisive?*

> **We extended the Conclusions with two additional bulleted points.**

*Some additional more specific comments include:*

*Line 21 and other instances - oxygen isotope stratigraphy and magnetostratigraphy are, strictly speaking, not 'dating methods' in themselves, although with appropriate calibration using geochronological methods have an obvious role in unravelling Earth history.*

> **We clarified in the text that we refer to dating and correlation techniques when using these terms.**

*Please be consistent in the spelling of racemization (either s or z but please be consistent throughout the text).*

> **Corrected.**

*Line 24 and other instances 'n' italic font*

> **Corrected.**

*Line 40 epimerization*

> **Changed.**

*Line 51 and elsewhere - do you mean calcium carbonate?*

> **Changed to calcium carbonate.**

*Line 62 'upper Quaternary' is not a stratigraphically recognised term - please be more specific*

> **Removed '*upper Quaternary*'.**

*Line 82 sample mass*

> **Changed.**

***Table 1*** *please indicate unit of measurement for temperature*

> **Corrected.**

***Line 119*** *correlated with*

> **Changed.**

***Line 149*** *mL and for microlitre later in the same paragraph*

> **Changed.**

***Line 186*** *high serine content - please quantify and explain in what sense.*

> **Clarified that this refers to subsamples with L-Ser /L-Asp ≥ 0.8.**

***Line 214*** *fossil age (sample is something that you have collected)*

> **Changed.**

***Figure 4 caption -*** *uncertainties rather than 'error bars' - they are not really an error, meaning something that is incorrect*

> **We agree, and we use the term "uncertainty" throughout in the text. For the figure captions, however, "error bar" is commonly used to refer to the depiction of uncertainties.**

***Line 249*** *compared with*

> **Changed.**

***Line 323*** *as above*

> **Changed.**

***Line 334*** *validity of this assumption?*

> **It is not known when *E. huxleyi* entered the Arctic Ocean – here we simply stated that the ages derived from the trend observed at the Greenland and Iceland seas could agree with the occurrence of *E. huxleyi* at these intervals, had this species entered the Arctic shortly after its evolutionary occurrence. We have altered the text to make this clearer.**

***Lines 341 to 345*** *Is this a manifestation of the kinetics of racemization and overall form of the extent of AAR with time?*

> **We are unsure of what the reviewer is asking. The ages of Arctic Ocean foraminifera are based on AAR global and GIS equations, which are empirical fits to D/L vs independent ages.**

***Line 364*** *but is this valid?*

> **How Arctic bottom water temperatures changed over time is poorly known, partly due to the chronological issues. We reiterated the findings from West et al. (2019) showing that temperature differences would need to be sustained at >4°C between sites to account for the D/L differences, which is unlikely.**

***Figure 9*** *benthic oxygen isotope curve - perhaps have times arrow reading to the righthand side of the page?*

The main feature of this figure is the age versus depth plot in the centre. For this plot, it is conventional for time zero to be positioned at the upper left surface.

**Line 391** *calcium carbonate.*

Changed to 'calcium carbonate'.

---

## Author Comment (AC3)

We appreciate the time the reviewer took to review the manuscript and thank the reviewer for their constructive comments. We have tried to address them all and feel that they have improved the manuscript.

The reviewer's comments are in black italics, followed by our responses in blue. A revised manuscript with tracked changes is also included.

*Anonymous Referee 3*

*This paper present new amino acid data from a series of sediment cores from the Arctic Ocean and the Nordic Seas. The new data include analyses of the planktonic species N. pachyderma and the benthic species C. wuellerstorfi and it is focused on the racemization reaction of Aspartic and Glutamic acid. The results are discussed in the context of the challenges in establishing robust chronologies for Arctic Ocean deep sea sediments. The paper is well written and has a sound scientific approach and should definitely be published. However, after reading the paper I think the authors should be a little bit stronger on their conclusion regarding the implications of their findings. To me it seems clear that the results suggest that either the established chronology is to young or the bottom waters have been warmer.*

*In the revision of the paper I would like the authors to consider expand with the following:*

*1. It is fine that the aa data generally support the correlation of cores based on density. However, merging all data on the depth scale of the ACEX core may introduce more scatter and uncertainty than needed.*

> We have considered this, however, there is insufficient geochronological control for the individual cores, so we are taking advantage of firm correlations to combine the data from multiple cores onto a single age scale. The illustrated correlation uses bulk density – because it is very straightforward to interpret. We also state, and have shown in past work, that this correlation is consistent with XRF based geochemistry (Zr, Ti, Mn, K), and grain size. References are provided for these works. We do not display all these correlation lines and data sets as it becomes rather untidy. We also acknowledge in the text that some scatter may arise from uncertainties in the correlation, but these are generally small (few cm) compared to the large scatter in the derived AAR ages. Understanding the origin of this scatter, and assessing whether it is greater than in other deep sea sites outside the Arctic, is a key step in future research.

*2. Present all Arctic Ocean cores with the basis for correlation also on depth scales.*

> We now include bulk density data displayed on individual core depths (Figure 2).

3. Expand the section with some hard data on the basis for the ACEX established chronology and expand the discussion of the aa results relative to the other methods used.

> A description of how the age model for the central Arctic cores was developed is on lines 130-145: "The currently accepted age model for the ACEX sedimentary sequence was developed using cyclostratigraphic analysis (O'Regan et al., 2008) and produced similar estimated Quaternary sedimentation rates as obtained by the decay of beryllium isotopes (Frank et al., 2008). The late Quaternary chronology (MIS 1 – 6) for ACEX included constraints from 14C dating, the correlation with nearby records AO96/12-1PC (Jakobsson et al., 2001) and PS2185 (Spielhagen et al., 2004), where MIS 5 was identified based on the occurrence of the calcareous nannofossil *Emiliania huxleyi* (Jakobsson et al., 2001), and further supported by results from optically stimulated luminescence dating of quartz grains (Jakobsson et al., 2003). The age model of core LOMROG07-PC04 is based on correlation with PS2185 (Hanslik et al., 2013)."

**Possibly this section was overlooked by the reviewer, but with more specific instructions on the kind of details that they want added, we would be happy to try and expand on this.**

*4. A review/figure/profile depicting the present main water masses in the Arctic Ocean may be useful. Perhaps some references to modelling work concerning possible temperature conditions in the Arctic ocean during the glacial stages may be useful-*

**We do not really know how water temperatures varied over the past. This is partly due to the chronological challenges associated with sediments from the central Arctic Ocean. We refer to Jones (2001) for a summary of water masses in the Arctic Ocean, and to the study by Cronin et al. (2012), who used numerical modelling to show that intermediate depth warming occurred during glacial conditions.**

*Miinor suggestions on text*

*30 ... bottom water temperatures may have been similar.*

**Changed.**

*33 ....models. Also a better understanding of temperature histories at the investigated sites and other possible environmental factors that may influence rasemisation rates in the central Arctic ocean, is needed.*

**Changed.**

*64 Bottom water of Atlantic and Pacific sites are presently generally a few degrees warmer than the Nordic Seas and Arctic ocean.*

**Added, "despite the cold bottom water in the central Arctic Ocean"**

*100 .....influence of Atlantic surface water, and...*

**Changed.**

*110 ...cores have been developed....*

**Changed.**

*141 ..with 21(?) samples...*

**A total of 95 stratigraphic depths were sampled resulting in 95 samples in total.**

*172 Where the reversal confirmed for both species at the same level?*

**Clarified by adding, "These include five samples of *N. pachyderma* and three *C. wuellerstorfi*. Of these, two samples contained sufficient tests of both species to analyse AAR, but only *C. wuellerstorfi* were stratigraphically reversed."**

*214 Considering what we know about past and present hydrography the samples from the Nordic Seas most likely have been exposed to the same water masses through time. Also the datapoints you have, seems not allow for establishing different pathways at the two sites. Suggest you make one polynomial fit but keep the coloring of points.*

**We consider it important to show that they independently follow similar trends.**

*249 This is surprising taken into account that the temperatures potentially have been lower than the "global" ocean.*

**Indeed.**

*Fig. 2*
*Suggest that the density data is plotted on individual core depths with correlation to ACEX indicated with lines.*

**We now include bulk density data displayed on individual core depths (Figure 2).**

*Fig. 5*
*It would have been nice to see the data for each core plotted on core depth in a separate figure.*

**We have provided this in Figure 2.**